# Added Value of Next Generation Sequencing in Characterizing the Evolution of HIV-1 Drug Resistance in Kenyan Youth

**DOI:** 10.3390/v15071416

**Published:** 2023-06-22

**Authors:** Vlad Novitsky, Winstone Nyandiko, Rachel Vreeman, Allison K. DeLong, Mark Howison, Akarsh Manne, Josephine Aluoch, Ashley Chory, Festus Sang, Celestine Ashimosi, Eslyne Jepkemboi, Millicent Orido, Joseph W. Hogan, Rami Kantor

**Affiliations:** 1Alpert Medical School, Brown University, Providence, RI 02912, USA; akarsh.manne@gmail.com; 2Academic Model Providing Access to Healthcare (AMPATH), Eldoret 30100, Kenya; nyandikom@yahoo.com (W.N.); rachel.vreeman@mssm.edu (R.V.); josteny@yahoo.com (J.A.); festero85@gmail.com (F.S.); celehaw@gmail.com (C.A.); eslynejepkemboi@gmail.com (E.J.); oridomillicent@yahoo.co.uk (M.O.); jwh@brown.edu (J.W.H.); 3College of Health Sciences, Moi University, Eldoret 30100, Kenya; 4Department of Global Health and Health System Design, Icahn School of Medicine at Mount Sinai, New York, NY 10029, USA; ashley.chory@mssm.edu; 5Arnhold Institute for Global Health, Icahn School of Medicine at Mount Sinai, New York, NY 10029, USA; 6School of Public Health, Brown University, Providence, RI 02912, USA; adelong@stat.brown.edu; 7Research Improving People’s Lives, Providence, RI 02903, USA; mark@howison.org

**Keywords:** HIV-1 drug resistance, children and adolescents, evolution of drug resistance, minority drug resistance variants, types of drug resistance evolution

## Abstract

Drug resistance remains a global challenge in children and adolescents living with HIV (CALWH). Characterizing resistance evolution, specifically using next generation sequencing (NGS) can potentially inform care, but remains understudied, particularly in antiretroviral therapy (ART)-experienced CALWH in resource-limited settings. We conducted reverse-transcriptase NGS and investigated short-and long-term resistance evolution and its predicted impact in a well-characterized cohort of Kenyan CALWH failing 1st-line ART and followed for up to ~8 years. Drug resistance mutation (DRM) evolution types were determined by NGS frequency changes over time, defined as evolving (up-trending and crossing the 20% NGS threshold), reverting (down-trending and crossing the 20% threshold) or other. Exploratory analyses assessed potential impacts of minority resistance variants on evolution. Evolution was detected in 93% of 42 participants, including 91% of 22 with short-term follow-up, 100% of 7 with long-term follow-up without regimen change, and 95% of 19 with long-term follow-up with regimen change. Evolving DRMs were identified in 60% and minority resistance variants evolved in 17%, with exploratory analysis suggesting greater rate of evolution of minority resistance variants under drug selection pressure and higher predicted drug resistance scores in the presence of minority DRMs. Despite high-level pre-existing resistance, NGS-based longitudinal follow-up of this small but unique cohort of Kenyan CALWH demonstrated continued DRM evolution, at times including low-level DRMs detected only by NGS, with predicted impact on care. NGS can inform better understanding of DRM evolution and dynamics and possibly improve care. The clinical significance of these findings should be further evaluated.

## 1. Introduction

Global scale-up of antiretroviral therapy (ART), received by 75% of 38.4 million people living with HIV/AIDS by the end of 2021, has saved millions of lives [1,2]. The emergence of drug resistance compromises ART effectiveness and remains a barrier to sustainable life-long ART [3,4,5]. Accumulation of drug resistance mutations (DRMs) over time can impact ART susceptibility, increase cross-resistance, compromise current treatment options and limit future ones [6,7]. While minority drug resistance variants, detectable by next generation sequencing (NGS), may be clinically relevant in certain circumstances, they are still not reported in routine clinical care and their clinical relevance is an existing research gap [8,9,10,11,12].

Children and adolescents living with HIV (CALWH) are more vulnerable than adults to developing treatment failure and drug resistance, particularly when infected perinatally, thus mandating life-long ART [13,14,15]. The vast majority (~90%) of CALWH failing ART in resource-limited settings have drug resistance [6,7,16,17,18,19,20,21,22]. Despite some emerging NGS data (e.g., Kemp et al., recently exploring drug resistance pathways in adults with HIV-1 subtype C failing 2nd-line ART, demonstrating extensive intra-host viral dynamics [23]), data on intra-host accumulation of DRMs over time, its impact, and the role of low-frequency DRMs in treatment experienced by CALWH in low-resource settings with diverse HIV-1 subtypes are limited [6,17,21,24]. 

We have been following a cohort of perinatally infected Kenyan CALWH failing first-line ART with extensive drug resistance and limited switch to second line ART, associated with long-term failure and mortality [25,26]. Findings emphasize the urgent need for interventions to sustain effective, life-long ART in this vulnerable population. Recently, we assessed the potential added value of using NGS over Sanger sequencing in detecting DRMs in this population. Despite good overall agreement between sequencing technologies at high NGS thresholds, even in this resistance-saturated cohort, 12% of participants had higher, potentially clinically relevant predicted resistance detected only by NGS, suggesting potential benefits of the more sensitive NGS over existing technology [27].

In this study, we characterize the evolution of drug resistance over time in this cohort of ART-experienced CALWH with non-B HIV-1 subtypes in a resource-limited setting, and further assess the potential value added by NGS in monitoring DRM evolution. By characterizing short- and long-term changes in DRM profiles under specific regimen selection pressure and examining their potential clinical effects, we hypothesize that minority resistance variants are more likely under drug selective pressure to evolve to levels that may impact clinical outcomes, possibly justifying the consideration of using NGS to detect such minority resistance variants early on.

## 2. Materials and Methods

### 2.1. Study Participants

CALWH were followed at AMPATH (Academic Model Providing Access to Healthcare) clinics in western Kenya, which provide free ART and associated services for >160,000 persons with HIV, including >10,000 youth [28,29], according to locally developed protocols per WHO guidelines. Treatment monitoring before 2016 was with CD4 testing at diagnosis and every 6 months thereafter, and viral load (VL) testing available on clinician suspicion of failure. In 2016, the Kenya Ministry of Health and AMPATH implemented routine VL testing and, since then, CALWH suspected as failing 1st-line regimens (consecutive VL > 1000 copies/mL or immunologic/clinical failure) were switched to standard 2nd-line protease inhibitor (PI)-based ART. Drug resistance data were not available at the time of switch. 

CALWH were enrolled at three timepoints (TPs): (1) in 2010–2013, as part of the Comprehensive Adherence Measurement for Pediatrics (CAMP) study, if they were perinatally infected with HIV, ≤14 years old, on or beginning 1st-line non-nucleoside reverse transcriptase inhibitor (NNRTI)-based regimens and in care at one of 4 AMPATH clinics; (2) in 2010–2013, as a ~3 month follow up; and (3) in 2016–2018, as part of the Resistance in a Pediatric Cohort (RESPECT) study. Participants were included in this study if their samples were successfully genotyped by NGS at least twice out of the three TPs. 

The study was approved by The Miriam Hospital and Mount Sinai Human Subjects Institutional Review Boards in the United States and the AMPATH Institutional Research Ethics Committee in Kenya.

### 2.2. Specimen Collection and Laboratory Methods

Whole blood was collected, and plasma was separated and stored (−80 °C) at each TP. CD4 and VL were tested routinely at sample collection and genotyping was attempted for all detectable (VL > 40 copies/mL) samples.

CD4 count and percent were tested using FACSCalibur flow cytometry (BD Biosystems, Erembodegem, Belgium). VL was tested using the Abbott Real Time HIV-1 assay on the m2000 system (Abbott Molecular, Inc., Des Plaines, IL, USA; lower detection limit 40 copies/mL). 

HIV-1 RNA was extracted from blood plasma by EZ1 Advanced XL (Qiagen, Germantown, MD) and using Qiagen EZ1 DSP Virus Kit (Qiagen GmbH, Hilden, Germany). HIV-1 RNA was converted to cDNA using SuperScript III First-Strand Synthesis System (Thermo Fisher Scientific, Waltham, MA, USA) and primer 3754 (5′ → 3′: CCAGGTGGCTTGCCAATACTCTGTCC, HXB2 nucleotide positions 3754–3779). Two rounds of PCR amplification were performed by using Phusion High-Fidelity DNA Polymerase (New England BioLabs, Ipswitch, MA, USA) and Platinum Taq DNA Polymerase High Fidelity (Thermo Fisher Scientific, Waltham, MA, USA) in the 1st (primers 1849, 5′ → 3′: GATGACAGCATGTCAGGGAG, HXB2 nucleotide positions 1827–1846 and 3754) and 2nd round (primers PRO-1, 5′ → 3′: CAGAGCCAACAGCCCCACCA, HXB2 nucleotide positions 2147–2166 and RT-21, 5′ →3′: CTGTATTTCTGCTATTAAGTCTTTTGATGGG, HXB2 nucleotide positions 3509–3539), respectively. Library preparation was performed using the Nextera DNA Library Prep Kit (Illumina) followed by NGS using the Illumina MiSeq platform. Generated sequences were quality controlled and processed using the bioinformatics pipeline hivmmer including assessment of amino acid NGS frequencies [30]. All samples in this study were processed using the same method for HIV-1 RNA extraction, amplification, NGS and bioinformatics pipeline. Resulting sequences (HXB2 nucleotide positions 2550–3509) encompassed the HIV-1 region encoding the first 320 codons of Reverse Transcriptase. Multiple sequence alignment was performed using *mafft* v7.450 [31]. HIV-1 subtyping was performed by using REGA v3 [32] with minor discrepancies resolved on a case-by-case basis.

### 2.3. Drug Resistance Analyses

To estimate short- and long-term drug resistance evolution, the time interval between visits ~3 months apart was defined as short-term, and between visits ~6 years apart as long-term. The frequency of each DRM was determined by NGS at each TP. Minority drug resistance variants were considered mutations when detected at NGS frequency 1–20%, not typically detected by traditional Sanger sequencing. DRMs were considered established at frequencies >20%, typically detected by Sanger sequencing. DRMs at NGS frequency <1%, typically undistinguishable from noise, were considered undetected. Thus, the term ‘≤20%’ used throughout the text is referring to both DRMs identified at NGS frequency 1–20% and undetected mutations, unless otherwise specified. A DRM was considered ‘detected’ if its NGS frequency was ≥1% at least at one TP.

Types of DRM evolution were defined based on the change (or lack of change) in NGS frequency of each mutation between two TPs (Table 1; Appendix A). An *evolving* DRM was defined as occurring at ≤20% at an early TP and >20% at a late TP; a *reverting* DRM was defined as >20% at an early TP and ≤20% at a late TP; DRMs with *increasing or decreasing* frequency were defined as mutations with a >5% frequency change within the 1–20% (minority resistance variants) or >20% (established DRMs) categories between an early and late TPs; and a *stable* DRM was defined as having a ≤5% frequency change within either category (minority or established DRMs).

Drug resistance evolution was assessed within study groups, which included short- and long-term time periods, and—within the latter—with and without ART regimen change. For each DRM evolution type, per-participant rate of evolution was computed as the proportion of individuals having the detected DRM. 

Drug resistance was assessed for nucleoside reverse transcriptase inhibitors (NRTIs) and NNRTIs. Predicted drug resistance scores and levels were assessed according to the penalty scores from the Stanford University HIV Drug Resistance Database HIVdb Program, ver. 8.9 [33,34]. The impact of evolving DRMs was evaluated on the basis of whether it affected the predicted resistance scores and/or changed predicted resistance levels. This analysis was performed both for medications taken at the time of sampling, and for potential future medications that were not previously taken—the NRTIs zidovudine (AZT) and tenofovir disoproxil fumarate (TDF), and the NNRTIs etravirine (ETR), rilpivirine (RPV) and doravirine (DOR).

### 2.4. Assessment of Potential Impact of Minority Resistance Variants on Mutation Evolution

In exploratory analyses, we examined associations between evolution of minority drug resistance variants and their potential clinical relevance. First, to address the hypothesis that established, minority or undetected drug resistance variants are more likely to evolve if the DRM has a positive penalty score towards the current regimen, we use regression analysis to quantify the association between DRM positive penalty scores and evolution in NGS frequency from early to later TPs. 

Second, we address the hypothesis that participants with minority drug resistance variants and positive penalty scores for the current regimen will have a lower predicted resistance score compared to those without minority drug resistance variants or with DRMs that have zero or negative penalty scores for current regimens. To accomplish this, we fit a regression model where change in the predicted resistance score is the dependent variable, and having a minority resistance variant with a non-zero penalty score is the independent variable. 

Associations are represented in terms of regression model coefficients, with bootstrap resampling used to generate confidence intervals. All analyses were implemented using R version 4.2.3 [35]. Full details are in the Appendix A.

## 3. Results

### 3.1. Study Participants

A total of 42 virologically unsuppressed CALWH had NGS sequences at ≥2 TPs and were included in the analyses. There were 22 short-term cases (mean interval between visits 2.8 months; IQR 2.8–2.9; range 2.6–3.1) and 26 long-term ones (mean interval between visits 5.6 years; IQR 4.6–6.6; range 3.9–7.8). Six participants had genotypes available at 3 TPs and were included in both short- and long-term evaluations. Over the short-term period, all participants remained on the same ART regimen. Over the long-term period, 7 participants remained on the same regimen, while 19 participants had a change of their ART regimen including 32% (6/19) staying on a NNRTI-based regimen and 68% (13/19) switching to a protease inhibitor (PI)-based regimen. 

The study cohort was predominantly (64%) male, and at enrollment had a mean age of 9 years (range 2–15 years), CD4 percent of 21, and VL of 41,547 copies/mL. ART regimens at TP1 included abacavir (ABC) or AZT or stavudine (D4T), lamivudine (3TC), and efavirenz (EFV) or nevirapine (NVP), taken for an average of 2.7 years. Participants harbored diverse HIV-1 subtypes, predominantly A1 and D (see Table 2 for further details and breakdown per study group).

### 3.2. Evolution of Drug Resistance 

Any evolution of DRMs between early and late TPs (i.e., any type of evolution examined other than stable mutations) was identified in 39/42 (93%) participants including 91% (20/22) in the short-term group, 100% (7/7) in the long-term group without regimen change, and 95% (18/19) in the long-term group with regimen change. Evolving DRMs (from ≤20% to >20%) were seen overall in 60% (25/42) of participants (10/22, 45% in the short-term group; 5/7, 71% in the long-term group without regimen change; and 14/19, 74% in the long-term group with regimen change; see Table 3 for details). Evolving DRMs from 1–20% to >20% occurred in 17% (7/42) of participants (5/22, 23% in the short-term group; 2/7, 29% in the long-term group without regimen change; and 3/19, 16% in the long-term group with regimen change).

Out of a total of 419 detected DRMs across all participant groups, evolving DRMs accounted for 12%, reverting DRMs 5%, established DRMs 44% (33% stable, 6% increasing and 4% decreasing), and minority DRMs 39% (30% stable, 4% increasing and 6% decreasing). Further details on DRM evolution types according to study groups and specific DRMs are provided in Table 4, Appendix A for NRTIs, and Table 5, Appendix A for NNRTIs.

Out of 190 detected NRTI mutations, evolving DRMs accounted for 15% (range 6–21% by study group), reverting 5% (range 3–9%), established 52% (range 38–70%; mostly stable), and minority resistance variants 28% (range 21–33%; mostly stable). Evolving DRMs from 1–20% to >20% occurred in 2/5 DRMs in the short-term group; 0/5 in the long-term group without regimen change; and 1/19 in the long-term group with regimen change. Breakdowns according to specific study groups and mutations are provided in Figure 1, Appendix A.

Out of 229 detected NNRTI mutations, evolving DRMs accounted for 10% (range 7–13% by study group), reverting for 4% (2–9%), established for 37% (range 33–39%; mostly stable), and minority resistance variants for 49% (45–52%; mostly stable). Evolving DRMs from 1–20% to >20% occurred in 3/8 DRMs in the short-term group; 3/4 in the long-term group without regimen change; and 3/14 in the long-term group with regimen change. Breakdowns according to specific study groups and mutations are provided in Figure 2, Appendix A.

### 3.3. Effect of Evolving DRMs on Predicted Resistance

Of the 25/42 participants with evolving DRMs, predicted resistance scores increased in 92% (23/25) and an escalation of predicted drug resistance level was found in 80% (20/25) of the participants, respectively. 

Of the 16/42 participants with evolving NRTI DRMs, evolution impacting a predicted score increase was seen in 88% (14/16) of them, including 75% (3/4) in the short-term group, 100% (3/3) in the long-term group without ART change, and 90% (9/10) in the long-term group with ART change (Figure 1). These changes resulted in an escalation of predicted drug resistance level in 69% (11/16) of participants, including 50% (2/4) in the short-term group, 67% (2/3) in the long-term group without ART change, and 80% (8/10) in the long-term group with ART change. The main evolving DRMs in participants without regimen change were related to ABC (e.g., L74V and Y155F, with an already-existing high level resistance to XTC at the earlier TP). 

Of the 20/42 participants with evolving NNRTI DRMs, evolution impacting a predicted score increase was seen in 85% (17/20) of cases, including 86% (6/7) in the short-term group, 100% (2/2) in the long-term group without ART change, and 83% (10/12) in the long-term group with ART change (Figure 2). These changes resulted in an escalation of predicted drug resistance level in 70% (14/20) of participants, including 86% (6/7) in the short-term group, 100% (2/2) in the long-term group without ART change, and 58% (7/12) in the long-term group with ART change. The main evolving DRM in participants without regimen change was the NVP-related Y181C. 

### 3.4. Analyses of Minority DRM Evolution

In the exploratory analyses to examine the evolution of minority drug resistance variants under a regimen that has positive penalty scores, we identified two NRTI (L74V and Y115F) and six NNRTI (A98G, G190A, H221Y, K103N, V108I, and Y181C) mutations in at least three participants with established DRMs at the later TP, and at least two participants with no penalty score for these DRMs at the later TP, who were candidates for further analysis. 

For NRTIs, participants taking at least one drug with an associated positive penalty score demonstrated higher increase in prevalence of both L74V and Y115F (compared to those without penalty score for those two DRMs), indicating more extensive minority variant DRM evolution due to the presence of a penalty score (Figure 3A). For both mutations, the impact of a penalty score on DRM prevalence at the later TP was larger in individuals with either minority resistance variants or established DRMs at TP1, but not among participants with undetected DRM at TP1.

For NNRTIs, participants taking at least one drug with an associated positive penalty score demonstrated higher increase in prevalence of Y181C if they had a minority (15% prevalence) or established (80% prevalence) DRM at TP1 (Figure 3B). Some minority drug resistance variants showed an empirical association with negative differences for the prevalence of K103N and G190A at higher TP1 prevalence (15 or 80%). However, the wide confidence intervals associated with these findings preclude drawing any definitive conclusion. 

Interestingly, we found 19 DRMs among 9/42 (21%) participants, all with ART regimen change, with emerging or evolving DRMs between TP1 and TP3, despite no apparent relevant drug pressure based on their ART regimen (Table 6). Of these 19 DRMs, 14 were evolving (7 from none to >20%; 3 from 1–20% to >20%), and 5 were established increasing (all with >5 percentage point increases).

In the second exploratory analysis, participants with at least one minority drug resistance variant with a positive penalty score for at least one drug in the current regimen in the early TP (compared to participants without minority drug resistance variants) had an elevated predicted resistance score for that drug in the later TP (Figure 4). In the short-term group, all analyzed antiretrovirals had larger positive changes in the resistance score at TP3 in the presence of at least one minority resistance variant at TP1. The largest differences were found for the two NNRTIs—EFV and NVP. In the long-term group, all but one antiretroviral (ABC) had larger positive changes in the resistance score at TP3 in the presence of at least one minority drug resistance variant at TP1. The antiretroviral for which minority drug resistance variants had the largest impact on the resistance score was AZT.

DRM profiles of the six participants who had genotypes available at all three TPs demonstrated multiple patterns of DRM evolution, following known drug-specific resistance pathways (Table 7). Evolving DRMs were detected in five of the six participants, NRTI DRMs in three and NNRTI DRMs in four, all at >20% frequencies, thus detectable by Sanger (highlighted in red in Table 7). Two of these participants had evolving DRMs to both antiretroviral classes. Some evolving DRMs (NRTI Y115F and NNRTI Y181C and H221Y) in three participants demonstrated long term sustainability evident from evolving at the second TP and maintaining presence over years until the third TP, again with >20% frequencies (highlighted by bold red in Table 7). Other evolving mutations were not sustainable and after appearing at the second TP were not detected at the third TP (e.g., NRTI L74V and NNRTI K103N in participant #2). Lastly, in three participants, minority drug resistance variants NRTI-Y115F and NNRTI- Y181C and K103N (not detectable by Sanger) evolved to established mutations within a few months, two of which were sustained for many years (highlighted in green in Table 7).

## 4. Discussion

In a unique ART-experienced cohort of CALWH with diverse HIV-1 subtypes in Kenya, longitudinally followed for up to ~8 years, we characterized drug resistance evolution and estimated the potential impact of minority drug resistance variants detected by NGS on clinical outcomes. Evolution of drug resistance was detected in almost every one of the 42 study participants, including evolution from minority to established resistance variants in some, even in this cohort with an already extensive drug resistance. This evolution led to a significant escalation of predicted drug resistance levels to both NRTIs and NNRTIs. Longitudinal follow-up and comprehensive analysis enabled identification of distinct types of DRM evolution and exploration of the impact of detecting minority drug resistance variants early on, which highlighted the potential added value of such an analysis. The new knowledge thus generated provides better and deeper understanding of drug resistance evolution, which might impact clinical care and deserves further study. 

The potential added value of this unique longitudinal analysis using NGS to detect evolution of minority drug resistance variants is demonstrated in several ways in this manuscript. First, in addition to the 60% of participants who had any type of DRM evolution, evolution from minority to established drug resistance variants was detected in 17%, suggesting that earlier consideration of minority DRMs might impact regimen selection even in this cohort with already high levels of drug resistance. Second, the changes in DRM frequencies that were detected by NGS were apparent in both the short and long-term studied groups, and in those with, but also perhaps surprisingly, in those without regimen changes, suggesting active replication and DRM selection in all circumstances, that should perhaps be monitored. This was also seen in the small but distinctive group of six participants with both short- and long-term follow-up opportunities, allowing the observation of a few evolving minority resistance variants that were sustained for many years. Third, exploratory analyses (limited by sample size) alluded to the possibility of higher evolution of minority resistance variants to both classes with existing relevant penalty scores, speculatively suggesting that the detection of minority resistance variants may indeed be relevant, as opposed to ‘noise’. Lastly, analyses enabled detection of evolution of DRMs despite no selective drug pressure, possibly due to three dimensional allosteric and/or functional compensation and covariation of mutations (not explored here), but also due to some still unknown mechanisms of drug resistance. Such exploration emphasizes the potential of NGS as a critical tool to gain new knowledge and identify avenues for further study of drug resistance dynamics. Overall, while recognizing the limitations of these data as outlined below, they suggest that incorporating individual and longitudinal NGS to monitor HIV drug resistance evolution in CALWH, and perhaps other populations, may be beneficial to support clinical care, and warrant further studies in larger cohorts and with more current antiretroviral medications. 

A comprehensive analysis in this study demonstrated continued, complex, extensive and diverse evolution of NRTI and NNRTI DRMs over time, even in a small sample size of 42 participants with heterogenous ART experience and already pre-existing high drug resistance. Participants experienced continuous evolution of drug resistance that was evident through dynamic changes of numerous DRMs spanning the entire range of NGS frequencies. Some DRM evolution was detected in 93% of participants, including 91% in the short-term group with no regimen change. Evolving DRMs were identified in 60% of participants when considering evolution from ≤20% to >20%, and in 17% when considering evolution from 1–20% to >20%. These findings highlight the need to closely monitor this vulnerable population using regular VL and drug resistance testing, and also suggest that early detection of DRMs at low NGS frequency might help improve care and facilitate rational regimen design, even in resource-limited settings with fewer treatment options.

Though our main focus was on development of mutations, we also detected reverting DRMs in 29% of participants (5% of all identified DRMs). Though little is known about reversion of DRMs in CALWH, this phenomenon may occur due to weakening drug selection pressure (e.g., sub-optimal adherence, treatment interruption or regimen change), or fitness costs [36,37,38,39]; however, its clinical significance is uncertain. In fact, most DRMs defined as reverting in this study had little or no effect on predicted resistance, likely due to other co-occurring DRMs. 

This study has several limitations. First, a small sample size limits analyses and renders some only exploratory in nature, though the analyses do investigate resistance evolution in a unique and vulnerable population. Second, the number of available follow-up timepoints is limited to two or three, possibly over-simplifying viral evolution. Third, the study uses only one (Stanford) resistance interpretation algorithm and also lacks functional testing, leading to only a prediction of clinical relevance, which requires further study. Fourth, use of older, mostly non-current ART regimens (though some with current specific medications) only provides a proof of concept, but mandates further investigation with more current regimens, different drug resistance barriers and resistance dynamics. Fifth, the impact of individual DRM viral fitness, VL, treatment interruptions or non-adherence were not available or considered. Lastly, NGS data were used without precise quantification of minority DRMs at low frequencies, such as in the primer ID approach [40,41], possibly resulting in some inaccurate mutation frequencies. Though considering relative comparisons (rather than absolute values) of NGS frequencies of the same DRMs in the same participants could alleviate this limitation, at least partially, the reported frequencies of <20% should be interpreted cautiously. 

## 5. Conclusions

In a small but unique cohort of Kenyan CALWH with pre-existing extensive HIV-1 drug resistance, we identified ongoing DRM evolution suggesting continued selection of minority and established viral variants under drug selection pressure. NGS and longitudinal follow-up of evolution of DRM frequencies can be informative for better understanding of the mechanisms and dynamics of HIV drug resistance evolution, and could play an important role in HIV monitoring and ART regimen selection. The clinical significance of the identified types of drug resistance evolution and whether they differ according to HIV-1 subtype remain to be evaluated in future studies.

## Figures and Tables

**Figure 1 viruses-15-01416-f001:**
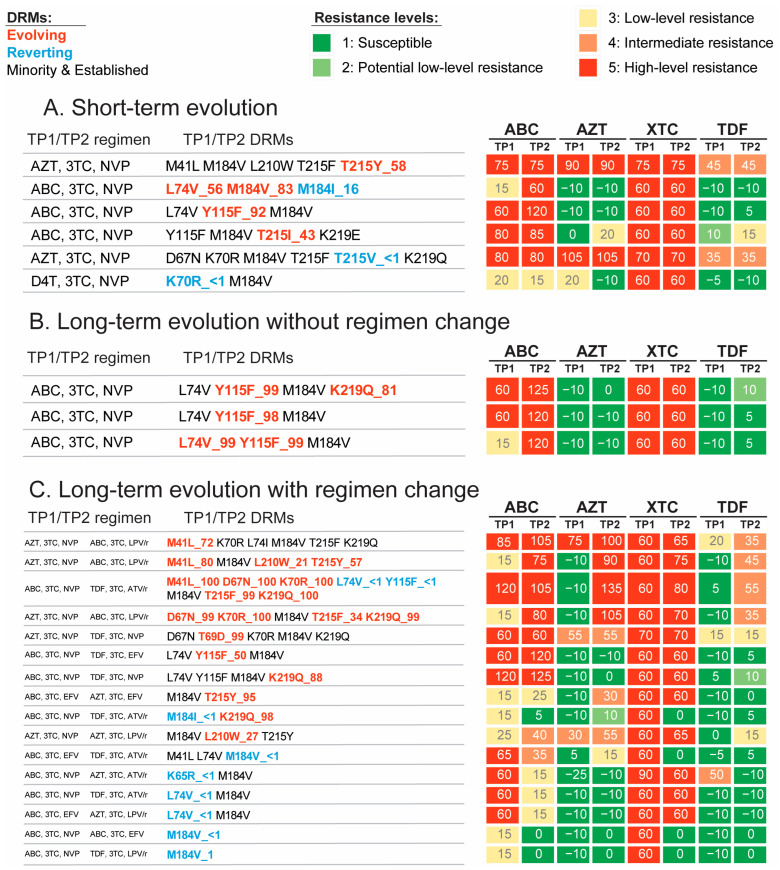
Participants with evolving and reverting NRTI DRMs. This figure presents NRTI DRM evolution and its predicted impact in study participants with short-term evolution (panel (**A**)), long-term evolution without regimen change (panel (**B**)), and long-term evolution with regimen change (panel (**C**)). Each row represents one study participant and includes regimens at earlier (TP1) and later (TP2) time points (first column); identified cumulative DRMs at TP1 and TP2 (second column); and predicted resistance scores (numbers) and levels (colors) at TP1 and TP2, depicted in blocks in the third column on the right, to NRTI drugs abacavir (ABC), zidovudine (AZT), lamivudine and emtricitabine (XTC) and tenofovir disoproxil fumarate (TDF). Identified DRMs are color-coded by evolutionary type according to the legend at the top left, including evolving DRMs in red; reverting DRMs in blue; and all other DRMs (including stable, and increasing/decreasing established/minority DRMs) in black. NGS mutation frequencies for the evolving and reverting DRMs are indicated by the number following the underscores (<1 = undetectable). Predicted resistance levels for each drug are presented as color-coded blocks according to the legend at the top right from green (susceptible) to high-level predicted resistance (red). In panel (**C**), the first column regimens are at TP1 and the second column regimens are at TP2.

**Figure 2 viruses-15-01416-f002:**
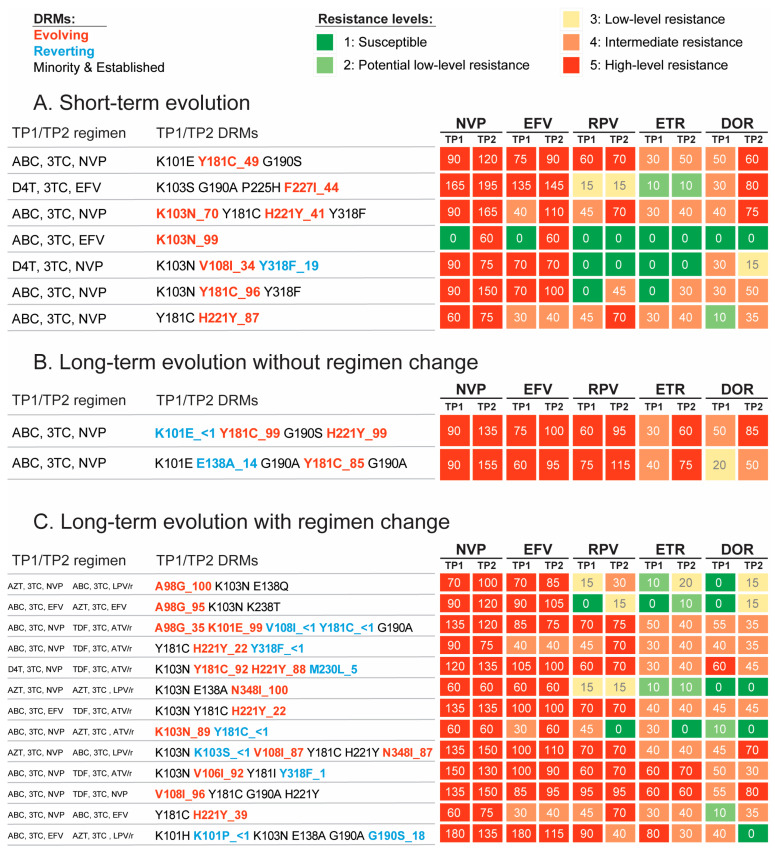
Participants with evolving and reverting NNRTI DRMs. This figure presents NNRTI DRM evolution and its predicted impact in study participants with short-term evolution (panel (**A**)), long-term evolution without regimen change (panel (**B**)), and long-term evolution with regimen change (panel (**C**)). For details and color-coding, see legend to Figure 1.

**Figure 3 viruses-15-01416-f003:**
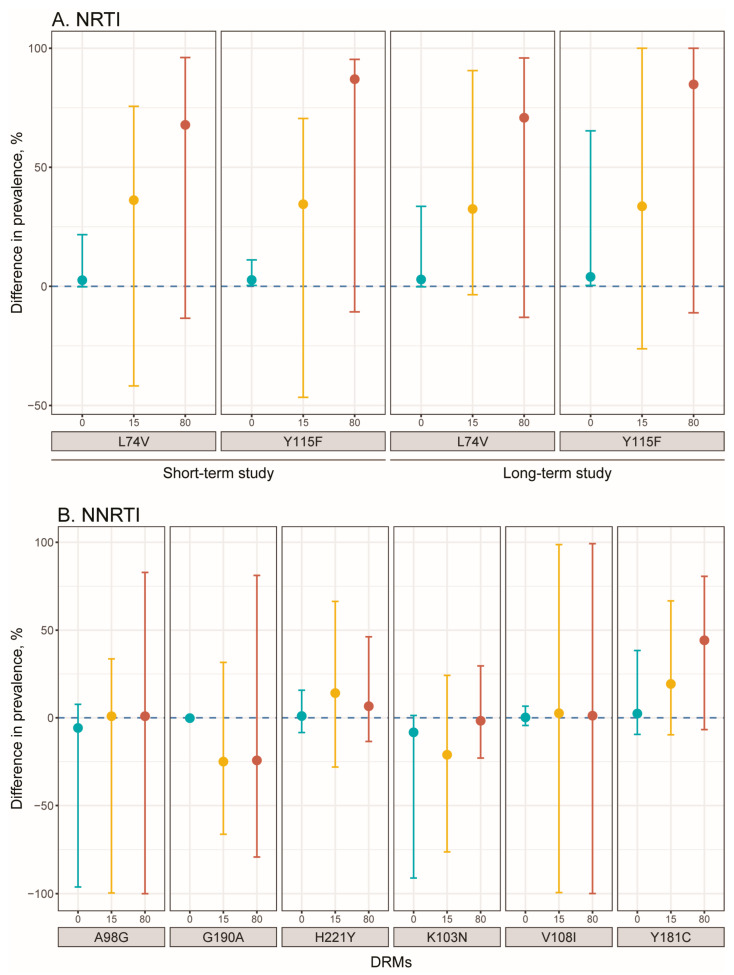
Difference in DRM prevalence at TP2 among participants taking a drug with penalty score in the short- and long-term studies (Exploratory Analysis 1). The Y-axis shows the estimated difference in TP2 DRM prevalence for participants taking at least one antiretroviral with a positive penalty score versus not taking an antiretroviral with a positive penalty score at TP2. Differences are presented for participants with TP1 DRM prevalence of 0, 15 or 80%. (**A**) NRTI DRMs at short- and long-term study (see details in Appendix A). (**B**) NNRTI DRMs at long-term study (see details in Appendix A).

**Figure 4 viruses-15-01416-f004:**
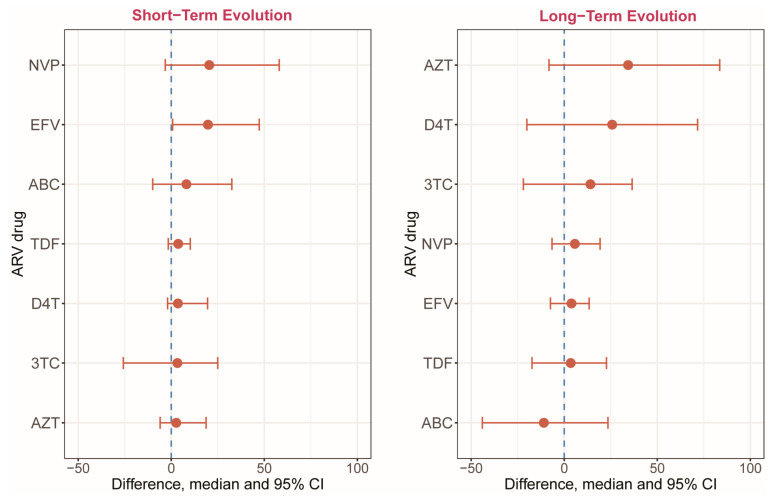
Difference in predicted resistance score at TP2 among participants with a minor drug resistance variant at TP1 with penalty score in the short- (left panel) and long-term (right panel) study (Exploratory Analysis 2). The X-axis shows the estimated difference at TP2 in resistance score for participants with at least one minority drug resistance variant at TP1 with positive penalty score versus those without a minority drug resistance variant with a positive penalty score. Contrasts are presented for each drug. The drugs are sorted by the median difference.

**Table 1 viruses-15-01416-t001:** Types of DRM evolution based on a change of NGS frequency over time.

Next Generation Sequencing Frequency		Type of DRM Evolution
Earlier Time Point	Later Time Point	
≤20%	>20%	Evolving DRM
>20%	≤20%	Reverting DRM
≤20%	≤20%; >5% increase	Minority DRM, increasing frequency
≤20%	≤20%; >5% decrease	Minority DRM, decreasing frequency
≤20%	≤20%; within 5%	Minority DRM, stable
>20%	>20%; >5% increase	Established DRM, increasing frequency
>20%	>20%; >5% decrease	Established DRM, decreasing frequency
>20%	>20%; within 5%	Established DRM, stable

DRM, drug resistance mutation; NGS, next generation sequencing.

**Table 2 viruses-15-01416-t002:** Cohort characteristics at enrollment according to time of follow up and ART regimen change.

Variable	Study Participants, Total, *n* = 42	Study Groups		
		Short-Term Evolution, *n* = 22	Long-Term Evolution without Regimen Change, *n* = 7	Long-Term Evolution with Regimen Change, *n* = 19
Gender (% females)	36	36	57	21
Age, mean years (range)	9 (2–15)	9 (6–15)	8 (2–13)	10 (5–13)
Mean CD4 count, cells/mm^3^ (range)	565 (42–2748)	423 (42–1062)	702 (394–1225)	684 (78–2748)
Mean CD4 percent (range)	21 (2–42)	18 (2–33)	25 (17–31)	23 (3–42)
Mean viral load, copies/mL (range)	41,547 (460–676,730)	67,342 (700–676,730)	7936 (1280–21,090)	41,725 (460–482,200)
Treatment failure, *n* (%)	39 (93%)	20 (91%)	7 (100%)	18 (95%)
**Regimens**				
ABC, 3TC, EFV/NVP; *n* (%)	22 (52%)	9 (41%)	5 (71%)	12 (63%)
AZT, 3TC, NVP; *n* (%)	13 (31%)	6 (27%)	2 (29%)	5 (26%)
D4T, 3TC, EFV/NVP; *n* (%)	7 (17%)	7 (32%)		2 (11%)
Mean time on ART, years (range)	2.7 (0.1–6.8)	2.5 (0.1–6)	2 (0.7–5.7)	2.9 (0.1–6.8)
**HIV-1 subtype/recombinant**				
A1; *n* (%)	26 (62%)	10 (45%)	4 (57%)	15 (79%)
A1, C; *n* (%)	2 (5%)	2 (9%)	1 (14%)	1 (5%)
A1, D; *n* (%)	3 (7%)	1 (5%)		2 (11%)
C; *n* (%)	3 (7%)	3 (14%)		
D; *n* (%)	8 (19%)	6 (27%)	2 (29%)	1 (5%)
**DRMs per participant**				
NRTI, mean (range)	2.5 (0–8)	2.9 (0–8)	1.9 (1–4)	2.3 (0–5)
NNRTI, mean (range)	2.2 (0–4)	2.1 (0–4)	2.3 (1–3)	2.2 (0–4)
NRTI and NNRTI, mean (range)	4.8 (0–12)	5 (0–12)	4.1 (2–6)	4.5 (0–7)

Abbreviations: 3TC, lamivudine; ABC, abacavir ART, antiretroviral therapy; AZT, zidovudine; D4T, stavudine; DRM, drug resistance mutation; EFV, efavirenz; NNRTI, non-nucleoside reverse transcriptase inhibitor; NRTI, nucleoside reverse transcriptase inhibitor; NVP, nevirapine.

**Table 3 viruses-15-01416-t003:** DRM evolution type according to participants and study groups.

Type of DRM Evolution		Number of Participants with DRMs, *n* (%)			
		Total, *n* = 42	Short Term, *n* = 22	Long Term with No Regimen Change, *n* = 7	Long Term with Regimen Change, *n* = 19
Evolving DRM		25 (60%)	10 (45%)	5 (71%)	14 (74%)
Reverting DRM		12 (29%)	4 (18%)	3 (43%)	9 (47%)
Minority DRM	Increasing	9 (21%)	3 (14%)	1 (14%)	6 (32%)
	Decreasing	17 (40%)	8 (36%)	3 (43%)	10 (53%)
	Stable	41 (98%)	22 (100%)	6 (86%)	19 (100%)
	Minority total	41 (98%)	22 (100%)	6 (86%)	19 (100%)
Established DRM	Increasing	18 (43%)	10 (45%)	4 (57%)	6 (32%)
	Decreasing	10 (24%)	5 (23%)	1 (14%)	7 (37%)
	Stable	37 (88%)	20 (91%)	6 (86%)	14 (74%)
	Established total	40 (95%)	21 (95%)	7 (100%)	18 (95%)
Any type except stable DRM		39 (93%)	20 (91%)	7 (100%)	18 (95%)

Abbreviations: DRM, drug resistance mutation.

**Table 4 viruses-15-01416-t004:** NRTI DRM evolution types according to mutations and study groups.

DRM Evolution Types		Study Groups								
		Short-Term Evolution, *n* = 22; ~3 Months			Long-Term Evolution, *n* = 26; ~6 Years					
					No Regimen Change, *n* = 7			Regimen Change, *n* = 19		
		Total per Group	Per Person	% *	Total per Group	Per Person	% *	Total per Group	Per Person	% *
Evolving DRM		5	0.23	5.6	5	0.71	20	19	1	20.7
Reverting DRM		3	0.14	3.4	1	0.14	4	8	0.42	8.7
Minority variant DRM	Increasing	1	0.05	1.1	0	0	0	7	0.37	7.6
	Decreasing	2	0.09	2.2	0	0	0	3	0.16	3.3
	Stable	16	0.73	18	7	1	28	20	1.05	21.7
	Total	19	0.87	21.3	7	1	28	30	1.58	32.6
Established DRM	Increasing	10	0.45	11.2	2	0.29	8	3	0.16	3.3
	Decreasing	3	0.14	3.4	0	0	0	9	0.47	9.8
	Stable	49	2.23	55.1	10	1.43	40	23	1.21	25
	Total	62	2.82	69.7	12	1.72	48	35	1.84	38.1
Total NRTI DRMs		89	4.06		25	3.57		92	4.84	

* Percent of the total number of identified DRMs per group. Abbreviations: DRM, drug resistance mutation; NRTI, nucleoside reverse transcriptase inhibitor.

**Table 5 viruses-15-01416-t005:** NNRTI DRM evolution types according to mutations and study groups.

DRM Evolution Types		Study Groups								
		Short-Term Evolution, *n* = 22; ~3 Months			Long-Term Evolution, *n* = 26; ~6 Years					
					No Regimen Change, *n* = 7			Regimen Change, *n* = 19		
		Total per Group	Per Person	% *	Total per Group	Per Person	% *	Total per Group	Per Person	% *
Evolving DRM		8	0.36	7.1	4	0.57	10.3	14	0.74	13.2
Reverting DRM		2	0.09	1.8	2	0.29	5.1	9	0.47	8.5
Minority variant DRM	Increasing	2	0.09	1.8	1	0.14	2.6	4	0.21	3.8
	Decreasing	11	0.5	9.7	4	0.57	10.3	10	0.53	9.4
	Stable	46	2.09	40.7	14	2	35.9	34	1.79	32.1
	Total minority variant DRM	59	2.68	52.2	19	2.71	48.8	48	2.53	45.3
Established DRM	Increasing	6	0.27	5.3	4	0.57	10.3	5	0.26	4.7
	Decreasing	4	0.18	3.5	1	0.14	2.6	4	0.21	3.8
	Stable	34	1.55	30.1	9	1.29	23.1	26	1.37	24.5
	Total established DRM	44	2	38.9	14	2	36	35	1.84	33
Total NNRTI DRMs		113	5.13		39	5.57		106	5.58	

* Percent of the total number of identified DRMs per group. Abbreviations: DRM, drug resistance mutation; NNRTI, non-nucleoside reverse transcriptase inhibitor.

**Table 6 viruses-15-01416-t006:** Long-term DRM evolution unrelated to drug selective pressure.

Patient ID	DRM	ARV Class	TP1 Regimen	TP2 Regimen	TP1 Penalty Score *	TP2 Penalty Score **	TP1 NGS, %	TP2 NGS, %
1	V108I	NNRTI	AZT, 3TC, NVP	ABC, 3TC, LPV/r	15	0	none	87
1	G190A	NNRTI	AZT, 3TC, NVP	ABC, 3TC, LPV/r	60	0	none	5
1	K103N	NNRTI	AZT, 3TC, NVP	ABC, 3TC, LPV/r	60	0	65	99
2	P225H	NNRTI	ABC, 3TC, NVP	AZT, 3TC, ATV/r	45	0	none	11
2	Y318F	NNRTI	AZT, 3TC, NVP	ABC, 3TC, LPV/r	30	0	14	88
2	K103N	NNRTI	ABC, 3TC, NVP	AZT, 3TC, ATV/r	60	0	none	89
3	T69D	NRTI	AZT, 3TC, NVP	TDF, 3TC, NVP	0	0	none	98
4	T69D	NRTI	ABC, 3TC, EFV	AZT, 3TC, LPV/r	10	0	79	99
5	A98G	NNRTI	AZT, 3TC, NVP	ABC, 3TC, LPV/r	30	0	none	100
5	Y318F	NNRTI	ABC, 3TC, NVP	AZT, 3TC, ATV/r	30	0	none	7
6	Y181C	NNRTI	D4T, 3TC, NVP	TDF, 3TC, ATV/r	60	0	15	92
6	K103N	NNRTI	D4T, 3TC, NVP	TDF, 3TC, ATV/r	60	0	57	75
6	H221Y	NNRTI	ABC, 3TC, NVP	TDF, 3TC, ATV/r	15	0	none	87
6	K103S	NNRTI	D4T, 3TC, NVP	TDF, 3TC, ATV/r	60	0	none	19
7	H221Y	NNRTI	D4T, 3TC, NVP	TDF, 3TC, ATV/r	15	0	12	88
8	K101E	NNRTI	ABC, 3TC, NVP	TDF, 3TC, ATV/r	30	0	none	99
8	A98G	NNRTI	ABC, 3TC, NVP	TDF, 3TC, ATV/r	30	0	none	35
8	G190A	NNRTI	ABC, 3TC, NVP	TDF, 3TC, ATV/r	60	0	66	99
9	K101H	NNRTI	ABC, 3TC, NVP	TDF, 3TC, ATV/r	60	0	70	100

* for any drug in the TP1 regimen; ** to any drug in the TP2 regimen. Abbreviations: 3TC, lamivudine; ABC, abacavir; ATV, atazanavir; AZT, zidovudine; D4T, stavudine; DRM, drug resistance mutation; EFV, efavirenz; LPV, lopinavir; NGS, next generation sequencing; NNRTI, non-nucleoside reverse transcriptase inhibitor; NRTI, nucleoside reverse transcriptase inhibitor; NVP, nevirapine; RTV, ritonavir.

**Table 7 viruses-15-01416-t007:** DRM profiles of six participants with available genotypes at three TPs.

ID	TP	Time	ART Regimen	NRTI DRM *	NNRTI DRM *
1	1	0	ABC, 3TC, NVP	L74V_35 M184V_93	A98G_7 K101E_32 V179D_1_L_4 Y181C_10_F_5 Y188F_6 G190S_99 H221Y_7 Y318F_9
	2	2.8 months	ABC, 3TC, NVP	L74V_99 M184V_95	V179L_4 Y181C_96_F_4 Y188F_4 G190S_99
	3	7 years	ABC, 3TC, NVP	K70N_1 L74V_100 Y115F_99 M184V_96 K219Q_81	**Y181C_99**Y188F_4 G190S_100 H221Y_99
2	1	0	ABC, 3TC, NVP	M184I_98	K103N_9 Y181C_97_S_1 Y188F_1 Y318F_24
	2	3 months	ABC, 3TC, NVP	L74V_56_I_13 V75A_2 Y115F_2 M184V_83_I_16 K219N_2	K101E_1 K103N_70 Y181C_98_S_1 H221Y_41 Y318F_25
	3	7.6 years	TDF, 3TC, ATV/r	K219Q_88	Y181C_84_S_2 **H221Y_87**
3	1	0	ABC, 3TC, NVP	M184V_99	Y181C_100
	2	2.6 months	ABC, 3TC, NVP	M184V_87	Y181C_88 H221Y_87
	3	7.2 years	ABC, 3TC, EFV	None	E138K_1 Y181C_38 **H221Y_39**
4	1	0	ABC, 3TC, NVP	L74V_82 Y115F_6 M184V_98	V179L_2 Y181C_97_S_2 H221Y_98 M230I_1
	2	2.9 months	ABC, 3TC, NVP	L74V_97 Y115F_92 F116Y_3 M184_V_98	V179L_2 Y181C_97_S_1 H221Y_99
	3	7.1 years	TDF, 3TC, EFV	L74I7_V_44 **Y115F_50**M184V_86	K103N_2 Y181C_97 H221Y_95
5	1	0	D4T, 3TC, NVP	V75S_2 M184V_59 T215F_43_I_11	K103N_57 Y181C_15 H221Y_12 M230L_48 Y318F_12
	2	3.1 months	D4T, 3TC, NVP	V75S_3 M184V_51 L210W_1 T215F_48	A98G_15 K103N_47 M230L_56 Y318F_5
	3	6.8 years	TDF, 3TC, ATV/r	V75S_1 M184V_95 T215F_93	K103N_75_S_19 Y181C_92 H221Y_88 M230L_5
6	1	0	D4T, 3TC, NVP	K65R_99 V75I_99 F116Y_99 Q151M_99 M184I_99	K103N_95_S_5 Y181C_98 Y318F_17
	2	2.8 months	D4T, 3TC, NVP	K65R_100 V75I_100 F116Y_99 Q151M_99 M184I_99	K103N_100 Y181C_98 Y318F_10
	3	5.7 years	ABC, 3TC, LPV/r	K65R_100 V75I_100 F116Y_98 Q151M_99 M184I_99	K103N_100 Y181C_97_S_1

* Number after underscore indicates % NGS frequency of the drug resistant amino acid at the specified amino acid position. Some positions have two drug resistance amino acids. Evolving mutations are highlighted in red; mutations evolving at the second TP and also identified at the third TP above 20%, are highlighted in bold red; minority resistance variants detected at 1–20% at an earlier TP and evolving to >20% at a later TP are highlighted in green. Abbreviations: 3TC, lamivudine; ABC, abacavir; ATV, atazanavir; D4T, stavudine; DRM, drug resistance mutation; EFV, efavirenz; LPV, lopinavir; NGS, next generation sequencing; NNRTI, non-nucleoside reverse transcriptase inhibitor; NRTI, nucleoside reverse transcriptase inhibitor; NVP, nevirapine; /r, ritonavir; TP, timepoint.

## Data Availability

Data collected for this study contains protected health information (PHI) of a vulnerable population of Kenyan children and adolescents, and cannot be made publicly available. Please contact Rami Kantor (rkantor@brown.edu) for data-related inquiries.

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
