# Peer review of "Added Value of Next Generation Sequencing in Characterizing the Evolution of HIV-1 Drug Resistance in Kenyan Youth"

_viruses, 2023, doi:10.3390/v15071416_

Round 1
Reviewer 1 Report
Next generation sequencing can detect mutations present at 1-20% of viral quasispecies as compared to Sanger sequencing deducing variants present at >20% of quasispecies. This paper by Novitsky et al. examines whether NGS is of added value in the analysis of evolution of HIV-1 drug resistance. The paper evaluated samples from 42 young Kenyan children and adolescents living with HIV (CALWH) at two or three timepoints over the 2010-2013 period.
Next generation sequencing can detect mutations present at 1-20% of viral quasispecies as compared to Sanger sequencing deducing variants present at >20% of quasispecies. This paper by Novitsky et al. examines whether NGS is of added value in the analysis of evolution of HIV-1 drug resistance. The paper evaluated samples from 42 young Kenyan children and adolescents living with HIV (CALWH) at two or three timepoints over the 2010-2013 period.
Next generation sequencing can detect mutations present at 1-20% of viral quasispecies as compared to Sanger sequencing deducing variants present at >20% of quasispecies. This paper by Novitsky et al. examines whether NGS is of added value in the analysis of evolution of HIV-1 drug resistance. The paper evaluated samples from 42 young Kenyan children and adolescents living with HIV (CALWH) at two or three timepoints over the 2010-2013 period. Major comments
There are several points that need to be addressed:
1. The premise that two timepoints reflect viral evolution over time (7 years) is overly simplistic.
2. The emergence and evolution of antiretroviral resistance to NRTIs and NNRTIs is a highly dynamic process, involving the interaction of multiple primary and secondary mutations. Single point mutations conferring NNRTI resistance, e.g., K103N confer high level resistance, as well as added fitness advantage That emergent resistance to NNRTI drug class persist at the second timepoint in the absence of non-nucleoside analogues, depicted in Table 6 is not unexpected. T69D, is also frequently transmitted mutation showing minimal impact on viral fitness.
3. Lines 60-62, The conclusion 12% of participants had higher, potentially clinically relevant predicted resistance detected only by NGS, can also be questioned. The authors used Stanford HIVdb program 8.9 to evaluate changes in resistance levels. This database does not consider quasispecies mixtures. A mutation present at 20% or 99% of quasispecies would be given the same penalty score. Moreover, each mutation is given an individual ranking that results in an overweighing of the accumulation of secondary mutations. The re-sensitization of viruses to NRTIs with M184I/V is given a -10 while the antagonism of K65R and NRTI pathways is given a -10. As such, Stanford reports are poor predictors of cumulative “true” phenotypic resistance.
4. Since Stanford is a poor predictor of cumulative resistance, Figure 1 should provide NGS frequencies at TP 1 and TP2 of added or removed mutations in Figure 1 and Figure 2.
5. It would be interesting to see if Geno2Pheno algorithms provide the same prediction. The 454 algorithm would select percentile mutation cut-offs for quasispecies mixtures.
6. Regarding Table 7, the discussion needs to address the differences in abacavir and TDF resistance pathways. Notably, no differences were noted that would have been detected by Sanger sequencing.
7. To pool evolving DRMs does not adequately address those mutations conferring low-, medium- and high-level resistance, their impact on viral fitness and their cumulative effects. Moreover, the study does not include VL that would identify treatment interruption or non-adherence. This should be emphasized in the discussion.
The English is fine.
Author Response
Major comments
- The premise that two timepoints reflect viral evolution over time (7 years) is overly simplistic.
Response: We agree that having additional, closer and more frequent, timepoints over longer periods of time would have enabled better characterization of viral evolution. To reflect this concern, we have added the following sentence to the ‘limitations’ paragraph in the Discussion: “Second, the number of available follow-up timepoints is limited to 2-3, possibly over-simplifying viral evolution.”.
- The emergence and evolution of antiretroviral resistance to NRTIs and NNRTIs is a highly dynamic process, involving the interaction of multiple primary and secondary mutations. Single point mutations conferring NNRTI resistance, e.g., K103N confer high level resistance, as well as added fitness advantage That emergent resistance to NNRTI drug class persist at the second timepoint in the absence of non-nucleoside analogues, depicted in Table 6 is not unexpected. T69D, is also frequently transmitted mutation showing minimal impact on viral fitness.
Response: We thank the Reviewer for this important comment, and agree with it. To address it we removed the word ‘unexpected’ from the manuscript, and (1) revised the Title of Table 6 to read: “Long-term DRM evolution unrelated to drug selective pressure.”; (2) revised the relevant sentence in the Results to read: “Interestingly, we found 19 DRMs among 9/42 (21%) participants, all with ART regimen change, with emerging or evolving DRMs between TP1 and TP3, despite no apparent relevant drug pressure based on their ART regimen (Table 6).”; and (3) revised the relevant Discussion sentence, to read: “Lastly, analyses enabled detection of evolution of DRMs despite no selective drug pressure, possibly due to three dimensional allosteric and/or functional compensation and covariation of mutations (not explored here), but also due to some still unknown mechanisms of drug resistance.”
- Lines 60-62, The conclusion 12% of participants had higher, potentially clinically relevant predicted resistance detected only by NGS, can also be questioned. The authors used Stanford HIVdb program 8.9 to evaluate changes in resistance levels. This database does not consider quasispecies mixtures. A mutation present at 20% or 99% of quasispecies would be given the same penalty score. Moreover, each mutation is given an individual ranking that results in an overweighing of the accumulation of secondary mutations. The re-sensitization of viruses to NRTIs with M184I/V is given a -10 while the antagonism of K65R and NRTI pathways is given a -10. As such, Stanford reports are poor predictors of cumulative “true” phenotypic resistance.
Response: We clarify that the statement in lines 60-62 is not related to this current paper, but rather summarizes the main results from our recently published paper. However, to address this comment, which might be relevant to the methods used in this paper as well, we have revised the relevant sentence in the ‘limitations paragraph’ of the Discussion, to include the fact that only one interpretation algorithm was used in this paper. We kept this addition general, since this is not the focus of this paper. The revise sentence now reads: “Third, methods include only one (Stanford) resistance interpretation algorithm and also lacks functional testing leading to only a prediction of clinical relevance, which mandate further study.”
- Since Stanford is a poor predictor of cumulative resistance, Figure 1 should provide NGS frequencies at TP 1 and TP2 of added or removed mutations in Figure 1 and Figure 2.
Response: As suggested, we added the actual NGS mutation frequencies at TP1 and TP2 to all evolving and reverting drug resistance mutations in Figures 1 and 2, and the Figure 1 legend was modified accordingly.
- It would be interesting to see if Geno2Pheno algorithms provide the same prediction. The 454 algorithm would select percentile mutation cut-offs for quasispecies mixtures.
Response: We agree with the Reviewer, however, as indicated in the comment above, comparing between drug resistance interpretation algorithms was not a focus of this paper and we are concerned that adding this focus would further complicate it. However, we hope that that the related modification above also addresses this comment, at least partially.
- Regarding Table 7, the discussion needs to address the differences in abacavir and TDF resistance pathways. Notably, no differences were noted that would have been detected by Sanger sequencing.
Response: Thank you for this comment. To address it we (1) incorporated the general concept that DRM evolution followed the drug-specific resistance pathways. We kept it general, since this was not the focus of this table, and we were concerned that discussing resistance pathways would further complicate this section. We revised the relevant sentence, to read: “DRM profiles of the six participants who had genotypes available at all three TPs demonstrated multiple patterns of DRM evolution, following known drug-specific resistance pathways (Table 7).”; and (2) revised the relevant section, indicate which type of evolution would and would not be detected by Sanger, which now reads: “Evolving DRMs were detected in five of the six participants, NRTI DRMs in three and NNRTI DRMs in four, all at >20% frequencies, thus detectable by Sanger (highlighted in red in Table 7). Two of these participants had evolving DRMs to both antiretroviral classes. Some evolving DRMs (NRTI Y115F and NNRTI Y181C and H221Y) in three participants demonstrated long term sustainability evident from evolving at the second TP and maintaining presence over years until the third TP, again with >20% frequencies (highlighted by bold red in Table 7). Other evolving mutations were not sustainable and after appearing at the second TP were not detected at the third TP (e.g., NRTI L74V and NNRTI K103N in participant #2). Lastly, in three participants minority drug resistance variants NRTI-Y115F and NNRTI- Y181C and K103N (not detectable by Sanger) evolved to established mutations within few months, two of which were sustained for many years (highlighted in green in Table 7).”
- To pool evolving DRMs does not adequately address those mutations conferring low-, medium- and high-level resistance, their impact on viral fitness and their cumulative effects. Moreover, the study does not include VL that would identify treatment interruption or non-adherence. This should be emphasized in the discussion.
Response: Thank you for this comment. We have designed Figures 1 and 2 to provide information on individual impact of mutations, their penalty scores to the individual drugs, and the level of predicted drug resistance they lead to. To fully address the comment, we have added some of these concerns to the limitations section of the Discussion, in a sentence that reads: “Fifth, the impact of individual DRM viral fitness, VL, treatment interruptions or non-adherence were not available or considered.”.
Reviewer 2 Report
In this study Novitsky et al. investigated the evolution of HIV-1 drug resistance by NGS in children and adolescents living with HIV. It is clear that high-throughput deep sequencing provides a more detailed view into accurate clinical drug resistance profiling; however, the clinical relevance of minority drug resistant variants is still under discussion. This manuscript presents important findings about the mechanism and dynamics of HIV drug resistance evolution in a longitudinal analysis, including minority drug resistance mutations. The results contribute to the better understanding of clinical implications of resistant variants with low abundance.
The manuscript is clearly written and well structured; however, there is some missing information and some points need to be corrected or revised to improve the study.
General concept comments:
- In the section 2.2. there is no information about nucleic acid isolation (applied method/kit). Please also indicate the primers used in cDNA synthesis (oligo dT/gene-specific primer) and in two rounds of PCR amplification. Did the authors use the same method for nucleic acid isolation, amplification (primers, PCR conditions), sequencing and bioinformatic analysis for all samples? If not, it may influence the frequency of drug resistance mutations, and in this case it should be mentioned in the discussion.
- Are there any evolutionary mechanisms which seem to be associated with certain HIV subtypes?
Specific comments:
- In the first column of Table 2. the word „mean” should be inserted into the row of CD4 count, CD4 percent and viral load (similar to the age in the third row). I suggest correcting „Time on ART” to „Average time on ART”. Also in the first column „n (%)” should be inserted next to the HIV subtypes.
- Please insert „range” before 3-9% in line 208 (similar to the other percentages in the sentence).
- I suggest using zero or minus in Table 4 instead of empty cells (in the row of increasing/decreasing minority variant and decreasing established DRM).
- I suggest using the same name for the first column of short-term evolution in Table 4 and Table 5 (Total/Per group).
- Please correct the NNRTI explanation in line 225 (“non-“ is missing).
Author Response
Comment: In this study Novitsky et al. investigated the evolution of HIV-1 drug resistance by NGS in children and adolescents living with HIV. It is clear that high-throughput deep sequencing provides a more detailed view into accurate clinical drug resistance profiling; however, the clinical relevance of minority drug resistant variants is still under discussion. This manuscript presents important findings about the mechanism and dynamics of HIV drug resistance evolution in a longitudinal analysis, including minority drug resistance mutations. The results contribute to the better understanding of clinical implications of resistant variants with low abundance.
The manuscript is clearly written and well structured; however, there is some missing information and some points need to be corrected or revised to improve the study.
Response: We thank the Reviewer for this summary and greatly appreciate his/her favorite review of our manuscript.
General concept comments:
Comment: In the section 2.2. there is no information about nucleic acid isolation (applied method/kit). Please also indicate the primers used in cDNA synthesis (oligo dT/gene-specific primer) and in two rounds of PCR amplification. Did the authors use the same method for nucleic acid isolation, amplification (primers, PCR conditions), sequencing and bioinformatic analysis for all samples? If not, it may influence the frequency of drug resistance mutations, and in this case it should be mentioned in the discussion.
Response: We added information to the ‘Specimen collection and laboratory methods’ Methods section on the method used for nucleic acid isolation, provided detailed information on cDNA synthesis and amplification primers, and confirmed that all samples in this study were processed using the same method for HIV-1 RNA extraction, amplification, NGS and bioinformatics pipeline.
Comment: Are there any evolutionary mechanisms which seem to be associated with certain HIV subtypes?
Response: This is an excellent and relevant question that we could not address in this study, probably due to the small study cohort. To address this comment and incorporate its relevance for future research, we have revised the final Conclusions sentence to read: “The clinical significance of the identified types of drug resistance evolution and whether they differ according to HIV-1 subtype remain to be evaluated in future studies.”.
Specific comments:
Comment: In the first column of Table 2. the word „mean” should be inserted into the row of CD4 count, CD4 percent and viral load (similar to the age in the third row). I suggest correcting „Time on ART” to „Average time on ART”. Also in the first column „n (%)” should be inserted next to the HIV subtypes.
Response: We introduced all suggested edits to Table 2.
Comment: Please insert „range” before 3-9% in line 208 (similar to the other percentages in the sentence).
Response: As suggested, we inserted “range” before “3-9%”.
Comment: I suggest using zero or minus in Table 4 instead of empty cells (in the row of increasing/decreasing minority variant and decreasing established DRM).
Response: As suggested, we added zeros to empty cells in Table 4.
Comment: I suggest using the same name for the first column of short-term evolution in Table 4 and Table 5(Total/Per group).
Response: As suggested, we modified column titles in Tables 4 and 5.
Comment: Please correct the NNRTI explanation in line 225 (“non-“ is missing).
Response: Thank you; this was corrected.
Round 2
Reviewer 1 Report
The Discussion and text were modified to address the majority of my comments and concerns.
The manuscript is now acceptable.